# YebC2 resolves ribosome stalling and increases fitness of cells lacking EF-P and the ABCF ATPase YfmR

Hye-Rim Hong◉⊙, Cassidy R. Prince◉⊙, Letian Wu◉⊙, Isabella N. Lin◉, Katrina Callan◉, Heather A. Feaga◉*

Department of Microbiology, Cornell University, Ithaca, New York, United States of America,

⊙ These authors contributed equally
* haf54@cornell.edu

**Data availability statement:** All data are in the manuscript and Supporting Information files (S1 Table and S1 Data).

**Funding:** This work was funded by NIGMS R35GM147049 to HAF. CRP was supported in part by a National Science Foundation Graduate Research Fellowship Program. The funders had no role in study design, data collection and

## Abstract

Ribosome stalling is a major source of cellular stress. Therefore, many specialized elongation factors help prevent ribosome stalling. One of the best characterized of these factors is EF-P, which prevents ribosome stalling at polyproline tracts and other difficult-to-translate sequences. Recent evidence suggests that other factors also facilitate translation of polyproline motifs. For example, YfmR was recently identified as a protein that prevents ribosome stalling at proline-containing sequences in the absence of EF-P. Here, we show that YebC2 (formerly YeeI) functions as a translation factor in *Bacillus subtilis* that resolves ribosome stalling at polyprolines. YebC2 associates with the ribosome, supporting a direct role for YebC2 in translation. Moreover, YebC2 can reduce ribosome stalling and support cellular fitness in the absence of EF-P and YfmR. Finally, we present evidence that YebC2 is evolutionarily distinct from previously characterized YebC-family transcription factors and demonstrate that these paralogs have distinct physiological roles in *B. subtilis*. Altogether our work identifies YebC2 as a translation factor that resolves ribosome stalling in *B. subtilis* and provides crucial insight into the relationship between YebC2, EF-P, and YfmR, three factors that prevent ribosome stalling at polyprolines.

## Author summary

Polyproline motifs are essential structural features of many proteins but are difficult for the ribosome to synthesize. EF-P reduces ribosome pausing at polyproline motifs. Here, we show that YebC2 (formerly YeeI) can also prevent ribosome stalling at polyprolines. YebC2 belongs to the YebC family of proteins which have been characterized as transcription factors. However, YebC2 is evolutionarily distinct from these factors and associates with the ribosome, indicating it plays a direct role in translation. YebC2 is important for fitness in the absence of EF-P. Moreover, YebC2 over-expression can rescue the severe fitness defect of cells lacking EF-P and the newly characterized anti-stalling factor YfmR. Our work suggests that EF-P, YfmR, and YebC2 act independently to prevent ribosome stalling and are important for maintaining cellular fitness.

analysis, decision to publish, or preparation of the manuscript.

**Competing interests:** The authors have declared that no competing interests exist.

## Introduction

Ribosomes catalyze peptide bond formation between amino acids to produce proteins. The polymerization rate is heavily influenced by the identity of the amino acids involved, with proline posing a special challenge since its side chain forms a rigid pyrrolidine loop that limits flexibility of the peptide backbone in the ribosomal exit tunnel [1,2]. EF-P was the first translation factor shown to resolve ribosome stalling at polyprolines and other difficult-to-translate sequences in bacteria [1–9]. EF-P interacts transiently with the ribosomal E site and then binds stably when tRNA^Pro is present in the P site [10,11] and promotes a favorable geometry of the polypeptide in the exit tunnel to facilitate peptide bond formation [1,2]. *efp* is essential in *Mycobacterium tuberculosis*, *Acinetobacter baumannii*, and *Neisseria meningitidis* [12–14]. In contrast, *efp* deletion from *Bacillus subtilis* causes sporulation and motility defects but does not cause a growth defect in standard lab conditions [15–19].

Recently, YfmR was identified as a protein that prevents ribosome stalling at polyproline tracts and Asp-Pro motifs in *B. subtilis,* and which is important for fitness in the absence of EF-P [20,21]. YfmR is a member of the ABCF family of ATPases that are widespread throughout bacteria and eukaryotes and have diverse roles in preventing ribosome stalling and mediating antibiotic resistance [22–27]. The *Escherichia coli* ortholog of YfmR, Uup, resolves ribosome stalling at polyprolines *in vitro* [28]. A recent structure of Uup bound to *E. coli* ribosomes reveals that it binds the ribosomal E site and makes contacts near the peptidyl-transferase center [29,30], suggesting that YfmR/Uup may promote peptide bond formation in a manner similar to EF-P. In support of this model, deletion of *yfmR* or *efp* does not result in a fitness defect in *B. subtilis*, whereas double deletion of both *yfmR* and *efp* results in a significant synthetic fitness defect [21].

The screen we used to identify YfmR also uncovered *yebC2* (formerly *yeeI*) as a gene that may be important for fitness in Δ*efp* cells. Consistent with this finding, a screen performed by Hummels and colleagues in 2019 identified *yebC2* (*yeeI*) as a gene whose over-expression could rescue the swarming motility defect of Δ*efp B. subtilis* cells [18]. YebC family proteins are annotated as transcription factors in bacteria since several of these proteins exhibit promoter binding activity and *yebC* deletion causes differential gene expression in *E. coli*, *Pseudomonas aeruginosa*, *Lactobacillus delbrueckii*, and *Borrelia burgdorferi* [31–34]. The human YebC homolog, TACO1, is localized to mitochondria where it is important for efficient translation of COX1 [35,36]. TACO1 was recently shown by mitoribosome profiling to prevent ribosome stalling at XPPX motifs and therefore accelerate translation of COX1 in human cells [37], and recent work by Ignatov and colleagues demonstrates that YebC in *Streptococcus pyogenes* (YebC_II) facilitates translation of polyproline motifs both *in vivo* and *in vitro* [38]. Altogether, these findings suggest that some YebC-family proteins play a role in translation.

Here, we show that *B. subtilis* YebC2 is a translation factor that prevents ribosome stalling at a polyproline tract and determine its genetic interaction with *efp* and *yfmR*. Depleting EF-P from Δ*yebC2* cells causes a significant fitness defect, and this defect is even more severe in Δ*yebC2*Δ*yfmR* cells. We find that Δ*yebC2* cells exhibit increased ribosome stalling at a polyproline track *in vivo* and that over-expression of YebC2 in Δ*efp* cells reduces ribosome stalling. We further show that YebC2 co-migrates with 70S ribosomes by sucrose density gradient ultracentrifugation, which suggests that YebC2 facilitates translation by acting directly on the ribosome. Finally, we present evidence that YebC2 proteins represent a class of translation factors that are evolutionarily distinct from the previously characterized YebC transcription factors.

## Results

### Deletion of *efp* and *yebC2* causes significant growth and translation defects

Previously, we investigated genetic interactions with *efp* using Tn-seq [20]. This screen predicted that *yfmR* is a gene that becomes important for fitness in the absence of EF-P, and we confirmed this result with CRISPRi [20]. Our Tn-seq screen also identified *yebC2* (*yeeI*) as a gene with significantly more transposon-insertions in the wild-type than in the Δ*efp* background suggesting that it may be important for fitness in the absence of *efp*. To test this, we deleted *yebC2* from Δ*efp* cells. Growth of Δ*efp*Δ*yebC2* is significantly impaired compared to Δ*efp* or Δ*yebC2* single deletions (Fig 1A). We complemented this growth defect by providing a single copy of *yebC2* integrated into the chromosome under the control of an IPTG-inducible promoter (Fig 1A). Moreover, Δ*efp*Δ*yebC2* cells exhibit a significant decrease in polysomes consistent with a defect in protein synthesis (Fig 1B). The decrease in actively translating ribosomes in the Δ*efp*Δ*yebC2* background was similar to what we and Takada and colleagues observed in Δ*efp*Δ*yfmR* cells [20,21].

### YebC2 over-expression rescues the synthetic fitness defect of Δ*efp*Δ*yfmR*

Previously, we found that deletion of *yfmR* in *B. subtilis* Δ*efp::mls* is lethal [20]. However, removal of the erythromycin resistance marker allows construction of the Δ*efp*Δ*yfmR* strain, but with a significant synthetic growth defect (Fig 2A) [21]. Therefore, we tested whether over-expression of YebC2 could rescue this synthetic defect. We expressed YebC2 under the control of an IPTG-inducible promoter in Δ*efp*Δ*yfmR* cells. At both 30°C and 37°C over-expression of YebC2 significantly improves fitness, as determined by growth in liquid media and colony size measurements (Figs 2A, 2B and S1). Rescue was especially pronounced at the lower temperature of 30°C (Fig 2A and 2B), consistent with previous observations that Δ*efp*Δ*yfmR* cells are especially sensitive to lower temperatures [21]. Additionally, we tested whether

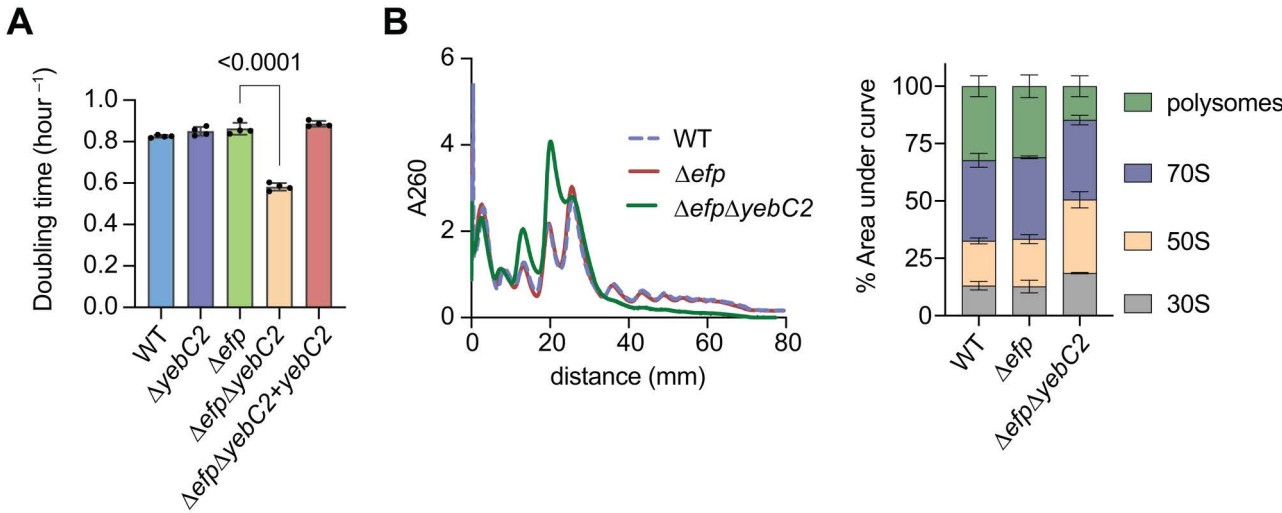

**Fig 1. Loss of *efp* and *yebC2* results in severe growth and fitness defects.** (A) Growth rates of Δ*efp* and Δ*efp*Δ*yebC2* in LB at 37°C. The growth defect is complemented by expressing YebC2 from an IPTG-inducible promoter (Δ*efp*Δ*yebC2* + *yebC2*). Error bars represent standard deviation of three independent experiments and p-values represent results of an unpaired *t*-test with Welch's correction. **(B)** Polysome profiles of wild-type, Δ*efp*, and Δ*efp*Δ*yebC2* strains. A representative of three independent experiments is shown. Quantification shows relative abundance of each ribosomal species as determined by area under each curve. Error bars represent standard deviation of three independent experiments.

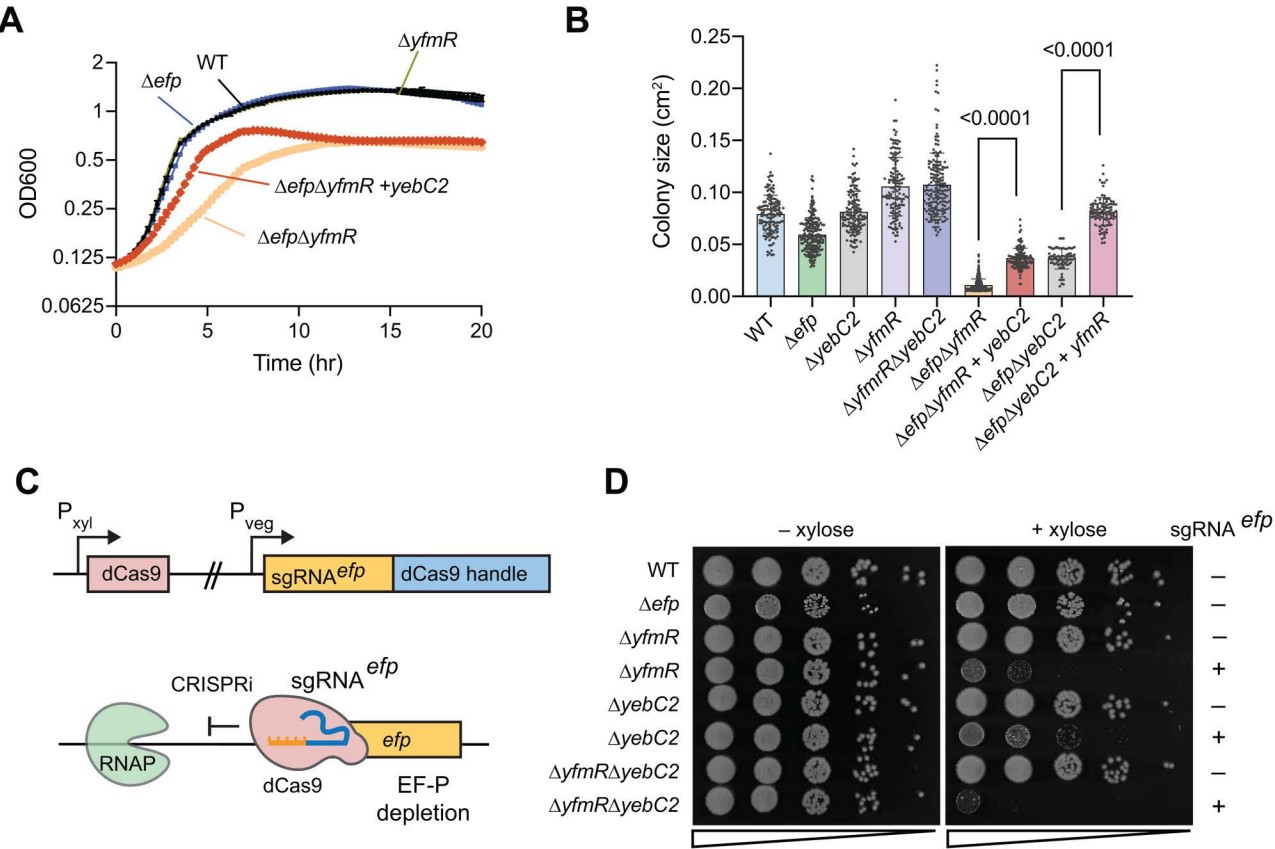

**Fig 2. Ectopic expression of YebC2 significantly increases fitness of Δ*efp*Δ*yfmR* cells. (A)** Growth in LB media at 30˚C of wild-type, Δ*efp*, Δ*efp*Δ*yfmR*, and Δ*efp*Δ*yfmR* expressing YebC2 from an IPTG-inducible promoter. Error bars represent standard deviation of 3 biological replicates. **(B)** Colony area measurements of indicated strains grown on LB plates for 48 hours at 30˚C. YfmR and YebC2 were expressed from an IPTG-inducible promoter. Error bars represent standard deviation and p-values represent results of an unpaired *t*-test with Welch's correction. Growth curves and colony sizing for these strains at 37˚C shows similar results and is available in S1 Fig. **(C)** Schematic of CRISPR interference used to deplete EF-P. Guide RNA targeting *efp* is expressed constitutively while deactivated Cas9 (dCas9) is expressed from a xylose-inducible promoter. Addition of xylose blocks transcription of *efp* thereby depleting EF-P. **(D)** Results of EF-P depletion from Δ*yfmR*, Δ*yebC2*, or Δ*yfmR*Δ*yebC2* double deletion. Culture was serially diluted and plated on LB with and without xylose to induce expression of dCas9. A representative of >3 independent experiments is shown.

YfmR over-expression could rescue growth of Δ*efp*Δ*yebC2* cells. Indeed, expression of YfmR in Δ*efp*Δ*yebC2* cells also rescued growth as determined by colony size measurements (Fig 2B). These data suggest that YebC2 supports cellular growth in the absence of EF-P and YfmR.

## YebC2 is important for cellular fitness in the absence of EF-P and YfmR

To further characterize the genetic interaction between *efp*, *yfmR,* and *yebC2*, we constructed a strain to deplete EF-P using CRISPR interference [39]. This strain expresses a guide RNA (sgRNA*efp*) that blocks transcription of *efp* when expressed alongside a deactivated Cas9 (dCas9) [39,40](Fig 2C). Consistent with our previous observations, depleting EF-P from Δ*yfmR* cells decreased colony formation by 3 orders of magnitude compared to when EF-P was not depleted (Fig 2D). EF-P depletion from Δ*yebC2* cells reduced colony formation by 2 orders of magnitude. Since depletion of EF-P from the Δ*yebC2* and Δ*yfmR* single deletions caused a significant fitness defect in both backgrounds, we next sought to deplete EF-P from a Δ*yebC2*Δ*yfmR* double deletion background. Δ*yebC2*Δ*yfmR* cells did not exhibit a fitness defect. However, when EF-P was depleted from Δ*yfmR*Δ*yebC2* cells, colony formation

decreased even more significantly than EF-P depletion from either of the single deletions (Fig 2D). These results demonstrate that YebC2 is important in cells lacking EF-P, and even more important in cells lacking both EF-P and YfmR. Moreover, since EF-P depletion from Δ*yfmR*Δ*yebC2* was more severe than depletion from either single mutant, and since over-expression of YebC2 or YfmR in the absence of the other two factors significantly rescues growth, we conclude that YebC2, YfmR, and EF-P can each independently support growth.

## YebC2 reduces ribosomal stalling at polyprolines

To determine whether YebC2 is important for preventing ribosome stalling at polyprolines, we used an *in vivo* stalling reporter encoding an N-terminal Flag tag for detection and five consecutive prolines mid-way through the protein sequence (Fig 3A). If ribosomes stall at the poly-proline tract a truncated stalled peptide is produced. Percent stalled peptide was determined by quantifying levels of stalled peptide divided by the sum of the stalled plus full-length peptide.

As expected, ribosome stalling is near undetectable in wild-type cells whereas significant stalling at the polyproline tract is observed in Δ*efp* cells (Fig 3A). Δ*yebC2* cells also exhibit levels of stalled peptide that are significantly higher than in wild-type cells (Fig 3A). The ribosome stalling observed in Δ*yebC2* cells (8 ± 1%) is not as high as in Δ*efp* cells (36 ± 4%), suggesting that EF-P is the main factor for preventing ribosome stalling at polyproline motif.

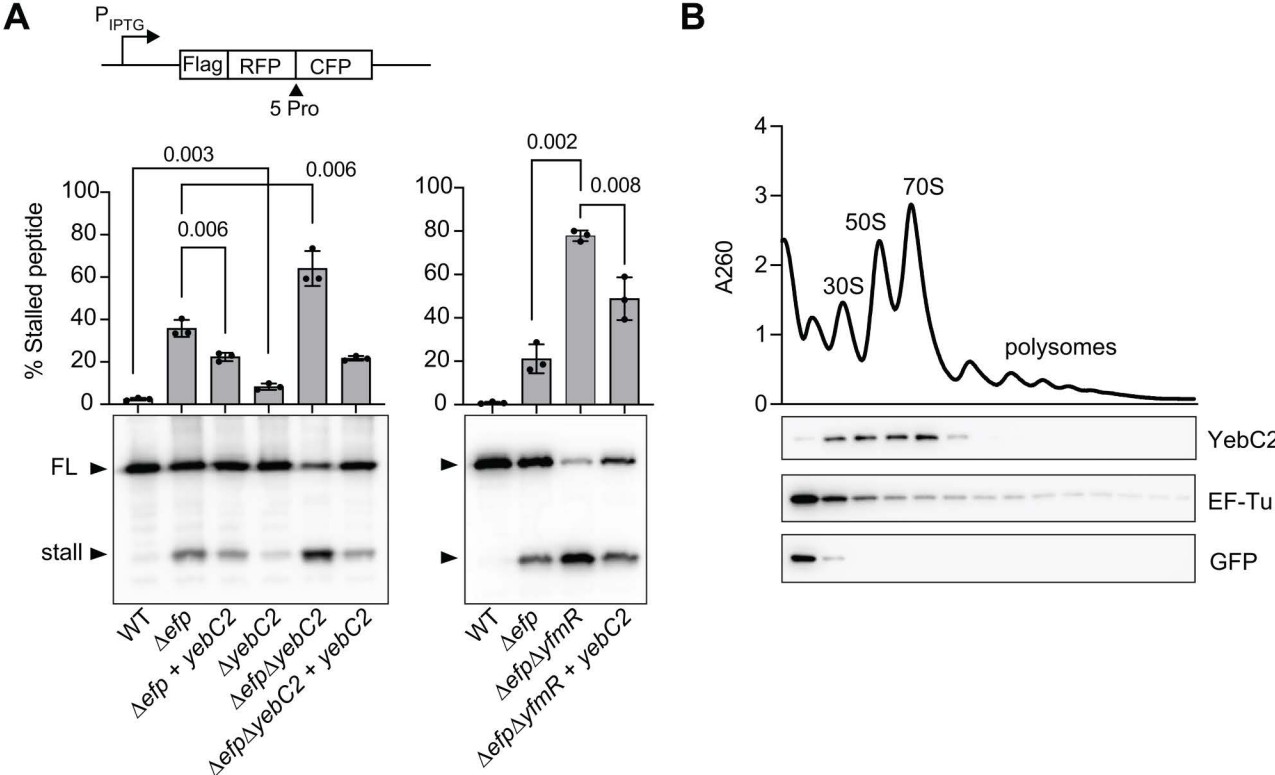

**Fig 3. YebC2 prevents ribosome stalling at a polyproline tract *in vivo* and associates with 70S ribosomes. (A)** A reporter encoding a Flag-tagged penta-proline tract was used to monitor ribosome stalling *in vivo*. Western blot shows levels of stalled and full-length (FL) peptide. Percent stalling is reported as level of stalled protein divided by the sum of stalled and full-length peptide. Error bars indicate standard deviation of 3 biological repli-cates. P-values report the results of an unpaired *t*-test. **(B)** Lysate from a strain expressing His-tagged YebC2 was resolved by sucrose density gradient ultracentrifugation. Fractions were probed with anti-His antibody or a polyclonal antibody raised against EF-Tu as a positive control for ribosome association. His-tagged GFP was used as a negative control for ribosome association.

Meanwhile, over-expression of YebC2 in Δ*efp* cells significantly reduces ribosome stalling (p = 0.0062). These results suggest that YebC2 prevents ribosome stalling at polyproline tracts.

Since both Δ*efp* and Δ*yebC2* single deletions exhibit significant stalled peptide, we next determined levels of stalled peptide in Δ*efp*Δ*yebC2* cells. These cells exhibit very high levels of stalled peptide (64 ±8%), significantly higher than either of the single deletions. Providing YebC2 under the control of an IPTG-inducible promoter in Δ*efp*Δ*yebC2* cells complemented this phenotype and reduced stalled peptide to levels lower than that of the Δ*efp* single deletion.

Since YebC2 over-expression in Δ*efp* cells significantly reduced ribosome stalling, we next asked whether YebC2 over-expression can also reduce ribosome stalling in cells lacking both EF-P and YfmR. As observed previously, loss of both EF-P and YfmR causes high levels of ribosome stalling [20] (Fig 3A). Ribosome stalling was significantly reduced when these cells were provided with YebC2 under the control of an IPTG-inducible promoter (Fig 3A). Decreased ribosome stalling when YebC2 is over-expressed in the absence of EF-P and YfmR further demonstrates that YebC2 can function independently of EF-P and YfmR to prevent ribosome stalling.

## YebC2 associates with ribosomes

To determine whether YebC2 interacts directly with the ribosome, we constructed a His-tagged version of YebC2 to monitor ribosome association. His-tagged YebC2 was functional, as evidenced by its ability to complement the impaired growth of Δ*efp*Δ*yebC2* cells (S2 Fig). Cells expressing His-tagged YebC2 were harvested in late exponential phase, and cell lysate was resolved by sucrose density gradient ultracentrifugation. We found that YebC2 co-migrates with ribosomes, including with 70S ribosomes (Fig 3B). In contrast, His-tagged GFP that served as a negative control for ribosome association was found only at the top of the gradient. Both 70S ribosomes and polysomes in sucrose density gradients contain actively translating ribosomes [41]. Although we observed YebC2 co-migration with 70S ribosomes, we did not detect YebC2 co-migration with polysomes, suggesting that either YebC2 does not interact with these ribosomes, or that the interaction is transient. Nevertheless, these results suggest that YebC2 exerts its anti-stalling activity by acting directly on the ribosome.

## YebC2 is evolutionarily distinct from YebC transcription factors

Many bacterial species, including *B. subtilis*, encode two YebC-family paralogs [42]. The *B. subtilis* YebC2 paralog is called YrbC. AlphaFold modeling of YebC2 and YrbC from *B. subtilis* predicts a high degree of structural similarity (Fig 4A). However, the results of our Tn-seq screen did not suggest a genetic interaction between *efp* and *yrbC* since we detected a similar number of transposon insertions in *yrbC* in Δ*efp* cells as in wild-type cells [20]. Consistent with the results of the transposon-insertion screen, we did not detect increased ribosome stalling at polyprolines in Δ*yrbC* cells, or in Δ*efp*Δ*yrbC* cells relative to Δ*efp* cells (Fig 4B). Additionally, over-expression of YrbC from the same promoter used to over-express YebC2 did not reduce ribosome stalling in the Δ*efp* strain (Fig 4B).

To determine the evolutionary relationship between the YebC paralogs we built a maximum likelihood tree based on the protein sequences of >15,000 YebC family proteins (Fig 4C). We found that the YebC paralogs that have experimental support for a role in transcription (YebC from *E. coli*, *L. delbrueckii*, *B. burgdorferi* and PmpR from *P. aeruginosa*) cluster together, while those that have a role in translation (*B. subtilis* and *S. pyogenes* YebC2 and *E. coli* YeeN) cluster separately (99.9% maximum likelihood bootstrap value) (Fig 4C). YebC2 from *B. subtilis, E. coli,* and *S. pyogenes* share a common ancestor exclusive of the YebC proteins that have been characterized as transcription factors (100% maximum likelihood

bootstrap value) (S3 Fig). Importantly, this clustering is not based on species phylogeny, since *B. subtilis* YrbC clusters with the YebC transcription factors.

### Residues that are important for the physiological function of YebC2 reside in Domain I

*B. subtilis* YebC2 and YrbC have high amino acid sequence identity (41%) (S3 and S4 Figs) and are predicted by AlphaFold to be structurally similar (Fig 4A). However, the results of our phylogenetic analysis (Fig 4C) and experiments with the polyproline stalling reporter (Fig 4B)

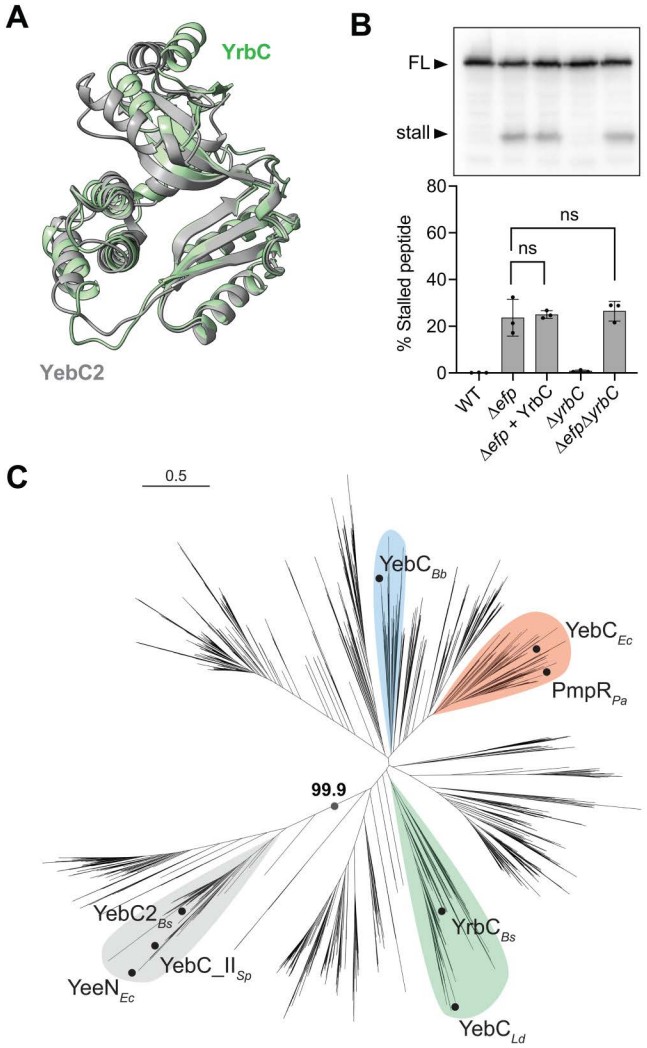

**Fig 4. YebC2 and YebC paralogs are structurally similar but evolutionarily distinct. (A)** AlphaFold model of the paralogs YebC2 (gray) and YrbC (green) from *Bacillus subtilis*. **(B)** Western blot showing levels of stalled and full-length (FL) peptide produced from a reporter for ribosome stalling at a penta-proline tract *in vivo*. Quantification reports the results of 3 biological replicates. Error bars represent standard deviation and p-values report the results of an unpaired *t*-test. **(C)** YebC2 proteins share a common ancestor exclusive of YebC-family transcription factors (99.9% maximum likelihood bootstrap value). Unrooted maximum-likelihood tree was built using all YebC family protein sequences detected in a database of >15,000 prokaryotic representative genomes. *Bs, Bacillus subtilis; Ld, Lactobacillus delbrueckii; Pa, Pseudomonas aeruginosa; Ec, Escherichia coli; Bb, Borrelia burgdorferi; Sp, Streptococcus pyogenes*. Clades containing proteins characterized in current literature are highlighted.

suggest that YebC2 and YrbC have distinct roles *in vivo*. To investigate potential differences between the two proteins we modeled their electrostatic potential using ChimeraX [43]. We found that while both proteins are highly negatively charged, YebC2 contains a region of positive charge on the surface of Domain I that is negatively charged in YrbC (Fig 5A, 5B, and 5C).

To test the contribution of Domain I to YebC2 function *in vivo*, we constructed strains expressing chimeric versions of YebC2 and YrbC. We replaced Domain I of YebC2 with Domain I of YrbC (YebC[YrbC-D1]) and we replaced Domain I of YrbC with Domain I of YebC2 (YrbC[YebC2-D1]). We expressed these chimeric proteins in the Δ*efp*Δ*yebC2* double deletion strain containing a reporter for polyproline stalling. As expected, expression of wild-type YebC2 complemented Δ*efp*Δ*yebC2* and reduced ribosome stalling below the levels of the Δ*efp* single

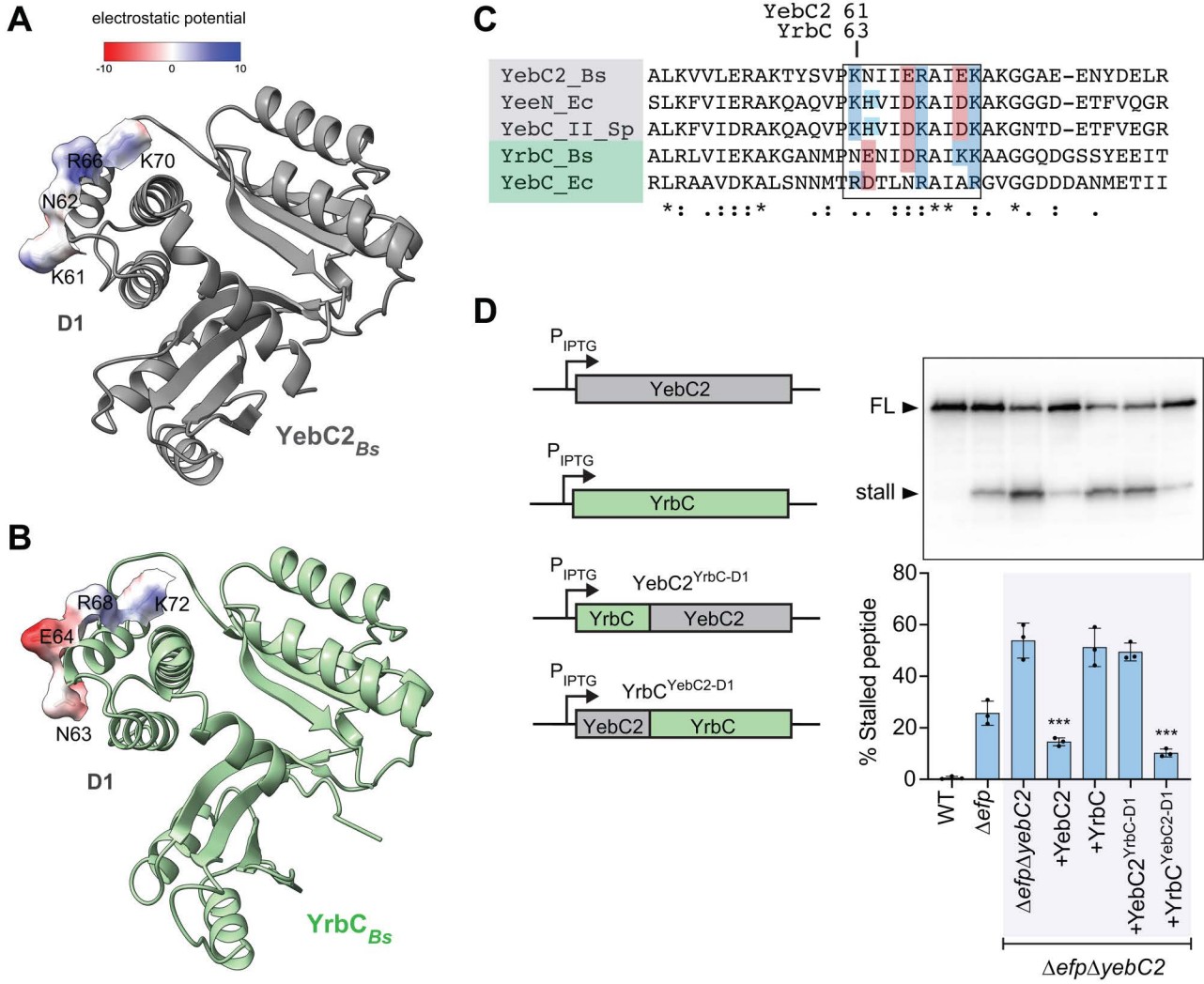

**Fig 5. Residues that are important for YebC2 function *in vivo* are in Domain I.** AlphaFold modeling of YebC2 (**A**) and YrbC (**B**) from *Bacillus subtilis*. Electrostatics are modeled with ChimeraX. (**C**) Amino acid sequence alignment from a portion of YebC2 and YebC paralogs highlighting difference in charges within a predicted surface region of Domain I. (**D**) Chimeric versions of YebC2 and YrbC as illustrated in the schematic were expressed in Δ*efp*Δ*yebC2* cells from an IPTG-inducible promoter. Flag-tagged *in vivo* reporter containing a penta-proline tract was used to determine which versions of the proteins could complement the ribosome stalling phenotype of Δ*efp*Δ*yebC2* cells. Quantification reports the results of 3 biological replicates. Error bars represent standard deviation. P-values report the results of an unpaired *t*-test that compares each of the indicated complementation strains to the Δ*efp*Δ*yebC2* uncomplemented strain. *** indicates p-value <0.001.

deletion (Fig 5D). In contrast, expressing YrbC from the same promoter did not reduce ribosome stalling. When we expressed YebC2 encoding Domain I from YrbC (YebC[YrbC-D1]), this chimeric version of the protein failed to complement the ribosome stalling phenotype of Δ*efp*Δ*yebC2*. However, when we expressed YrbC encoding Domain I of YebC2 (YrbC[YebC2-D1]), this chimeric version complemented the Δ*efp*Δ*yebC2* ribosome stalling phenotype to the same level as wild-type YebC2. These results suggest that residues that are important for YebC2 function reside in Domain I.

## YebC proteins are widely distributed in bacteria while YebC2 proteins are more restricted

Having determined that YebC and YebC2 proteins are evolutionarily distinct, we next determined the conservation of these paralogs across the bacterial domain (Fig 6) (S1 Table). 87% of the >15,000 bacterial genomes we surveyed encode at least one YebC-family protein (either YebC or YebC2), consistent with its likely presence in the common ancestor of bacteria [42]. YebC is much more widely distributed and highly conserved than YebC2. We detected YebC in 80% of our surveyed genomes and YebC2 in only 13%. YebC2 was mainly restricted to Firmicutes (Bacillota) and Gammaproteobacteria. Interestingly, organisms that encode YebC2 were more likely to lack the YebC paralog. Of the taxa encoding YebC2 only 34% encoded YebC. Interestingly, some organisms encode up to 3 YebC paralogs and up to 2 YebC2 paralogs (Fig 6) (S1 Table). The broad conservation of YebC-family proteins suggests that they impart a strong selective advantage in the organisms that encode them.

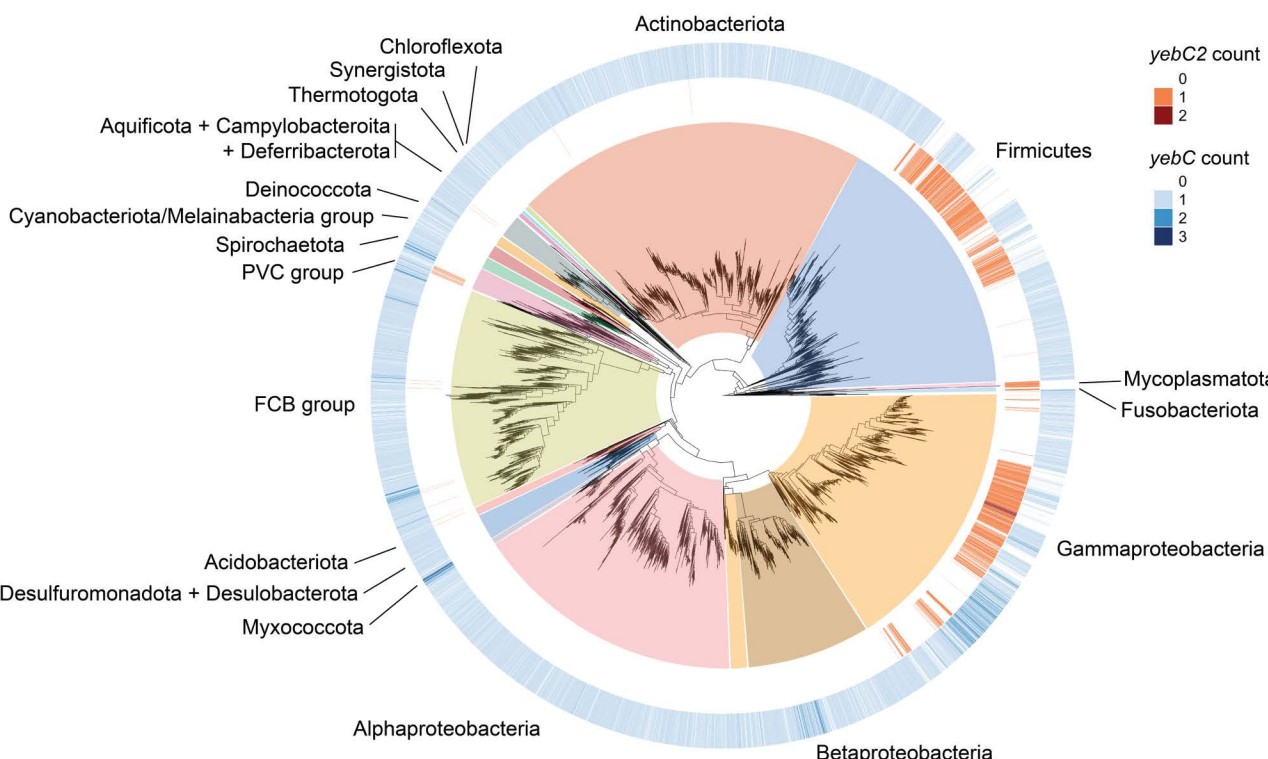

**Fig 6. Distribution of YebC-family paralogs in bacteria.** A midpoint rooted 16S maximum-likelihood phylogenetic tree of species in the bacterial domain, indicating the number of *yebC2* or *yebC* genes in each genome. YebC2 paralogs are most well-conserved in Firmicutes (Bacillota), and Gammaproteobacteria whereas YebC paralogs are widely distributed across most bacterial phyla. 87% of surveyed genomes encode at least one YebC-family protein.

## Discussion

Here we show that YebC2 is a ribosome-associated protein that reduces ribosome stalling at a penta-proline tract *in vivo* (Fig 3). These findings are in complete agreement with the elegant work of Brischigliaro and Krüger and colleagues and Ignatov and colleagues which recently determined a similar role for human TACO1 and *S. pyogenes* YebC_II [37,38]. Our work further shows that simultaneous loss of YebC2, EF-P, and YfmR greatly reduces the viability of the bacterium *B. subtilis* (Figs 1 and 2), and that YebC2 can reduce ribosome stalling in the absence of EF-P and YfmR (Fig 3). We also detect YebC2 associated with ribosomes, including 70S ribosomes, suggesting that YebC2 may function in translation elongation (Fig 3B). Altogether, this work contributes to a more complete understanding of the various factors that prevent ribosome stalling at polyproline tracts.

Our data support a model in which EF-P, YfmR, and YebC2 independently prevent ribosome stalling and support cellular fitness. An independent role for YebC2 is demonstrated by its ability to prevent ribosome stalling on a polyproline tract in cells lacking both EF-P and YfmR (Fig 3). If YebC2 were dependent on either EF-P or YfmR for its activity, it would be unable to rescue growth or prevent ribosome stalling in the Δ*efp*Δ*yfmR* background. In agreement with these results, EF-P depletion from Δ*yfmR*Δ*yebC2* cells reduces viability more than EF-P depletion from either the Δ*yfmR* or Δ*yebC2* single deletions (Fig 2). Thus, loss of all three factors is more detrimental than loss of any two factors which further suggests that these factors have some redundant function and that the presence of at least one factor is important for fitness.

How does YebC2 reduce ribosome stalling? Using a reporter for ribosome stalling at a polyproline tract, we observe that Δ*efp*Δ*yebC2* and Δ*efp*Δ*yfmR* cells exhibit increased levels of stalled peptide accompanied by decreased levels of full-length peptide (Fig 3A). In both strains, over-expression of YebC2 decreases levels of truncated peptide while increasing levels of full-length peptide (Fig 3A). These observations suggest that YebC2 may directly promote translation of the full-length peptide. Consistent with this possibility, Ignatov and colleagues found that YebC_II from *S. pyogenes* likely makes contacts with the base of Helix 89 of the large ribosomal subunit near the region where the acceptor stems of the A- and P-site tRNAs come into proximity for peptidyl transfer [38]. A structure of YebC2 bound to the ribosome is necessary to show the precise role YebC2 in enhancing translation at polyprolines.

In *B. subtilis*, ribosomes that stall upstream of the 3' end of the mRNA are split from the mRNA by MutS2 and the stalled peptide remains bound to the P-site tRNA and obstructs the exit tunnel of the large subunit [44,45]. The stalled peptide is targeted for degradation through the template-free addition of an alanine tag by RqcH [46–49]. The stability of the truncated peptide produced from ribosome stalling at polyprolines suggests that addition of alanine to the polyproline tract may pose a special challenge for RqcH. Interestingly, the eukaryotic EF-P homolog, eIF5A, facilitates CAT-tailing by the RqcH homolog Rqc2 [50]. Therefore, along with promoting translation of full-length peptide, it remains an exciting possibility that YebC2 promotes alanine addition by RqcH. These possibilities are not mutually exclusive, both involve a role for YebC2 in promoting peptidyl-transfer, and the co-migration of YebC2 with 70S and 50S ribosomes that we observe by sucrose density gradient ultracentrifugation is consistent with either of these possibilities (Fig 3B).

Although we observe anti-stalling activity for EF-P, YfmR, and YebC2 on a penta-proline reporter, it is likely these factors prevent ribosome stalling at sequences that extend beyond prolines. For example, YfmR also prevents ribosome stalling on polyacidic residues [21]. Meanwhile, EF-P promotes peptide bond formation at other difficult-to-translate sequences [51,52]. EF-P likely plays a role in formation of the first peptide bond since it recognizes both tRNA[Pro] and initiating tRNA[fMet] in the P site [11,53]. Moreover, EF-P promotes peptide bond formation

between initiating formyl-methionine and the second amino acid and helps maintain the reading frame during early elongation [54–56]. YfmR may also participate in early elongation since YfmR depletion in Δ*efp* cells causes increased association of initiator tRNA with stalled ribosomes [20]. Finally, there is also evidence that YebC2 plays a role at non-proline encoding sequences since deletion of the YebC2 ortholog in yeast causes a more general defect in protein synthesis, with reduced overall synthesis of mitochondrial-localized reporters [57].

YebC family proteins are widely annotated as transcription factors [31–34,58,59]. By analyzing YebC family protein sequences, we found that these proteins cluster into divergent clades in agreement with experimental evidence supporting their roles in either transcription or translation (Fig 4). Since *B. subtilis* encodes both YebC2 and YebC (YrbC) we investigated the physiological role of each protein *in vivo*. We found that while YebC2 over-expression from an IPTG-inducible promoter could reduce levels of ribosome stalling at a polyproline tract in Δ*efp* cells (Fig 3A), YrbC expression from the same promoter did not reduce stalled peptide levels (Fig 4B). Moreover, whereas we detect significant levels of stalled peptide in Δ*yebC2* cells, we do not detect significant stalling at the same polyproline tract in Δ*yrbC* cells (Fig 4B). Interestingly, *E. coli* YebC resolves ribosome stalling at a penta-proline tract *in vitro* [38] despite its reported role in transcription [34]. Since *E. coli* YebC clades with characterized YebC/YrbC transcription factors (Fig 4C), it is possible that *B. subtilis* YrbC also reduces ribosome stalling *in vitro*. However, our data suggest that YrbC does not have a significant physiological role in reducing ribosome stalling at a polyproline tract *in vivo*.

YebC2 is conserved primarily within Firmicutes (Bacillota) and Gammaproteobacteria while YebC is broadly conserved and was likely present in the common ancestor of bacteria (Fig 6). The retention of both YebC and YebC2 paralogs in many taxa further supports a model in which these proteins impart unique selective advantages due to independent physiological roles. We also note that many taxa encode more than one YebC2 and up to three YebC paralogs per genome (Fig 6). A similar observation has been made recently for EF-P, in which EF-P paralogs (EfpL) have been identified in approximately 12% of bacterial genomes [60]. Ribosome profiling revealed that EF-P and EfpL have both overlapping and non-overlapping substrate specificities, and are subjected to different modes of post-translational modification to regulate their activities [60]. Future work is necessary to determine the precise substrates and physiological roles of the YebC2 paralogs in diverse species.

## Materials and Methods

### Strains and media

Strains were derived from *B. subtilis* 168 trpC2 and are listed in Table 1. Single deletions were obtained from the BKK collection [63] and moved into the lab's 168 trpC2 strain by natural transformation. The kanamycin resistance cassette was excised to make clean deletions using pDR244 [63]. *B. subtilis* strains were cultured in LB and supplemented with antibiotics at final concentrations of 100 μg/mL spectinomycin, 1x MLS (1 μg/mL erythromycin and 25 μg/mL lincomycin), or 5 μg/mL chloramphenicol. *E. coli* DH5alpha strains were cultured in LB with 100 μg/mL ampicillin.

### Complementation of *yebC2* and *yfmR*

Primers are listed in Table 1. *yebC2* was amplified from the wild-type *B. subtilis* 1772 WT 168 trpC2 genomic DNA using primers HRH155 and HRH156 which contain 22 bp of homology to pDR111. Primers HRH157 and HRH158 were used to amplify *yfmR*. The resulting fragments were cloned by Gibson assembly into pDR111 cut with HindIII and SphI. The resulting

**Table 1. Strains, plasmids, and primers.**

| Strain (strain number) | Description | Source |
|---|---|---|
| HAF1 | 168 trpC2 *B. subtilis* wild type | [61] |
| HAF242 | 168 trpC2 Δefp::kan | [19] |
| HAF450 | 168 trpC2 ΔyebC2::kan | This study |
| HAF451 | 168 trpC2 ΔyfmR::kan | This study |
| HRH575 | 168 trpC2 Δefp | This study |
| HRH802 | 168 trpC2 ΔyebC2 | This study |
| HRH804 | 168 trpC2 ΔyfmR | This study |
| HAF519 | 168 trpC2 Δefp ΔyebC2::kan | This study |
| HAF521 | 168 trpC2 Δefp ΔyfmR::kan | This study |
| HRH1132 | 168 trpC2 ΔyebC2::kan ΔyfmR | This study |
| HAF518 | 168 trpC2 Δefp ΔyebC2::kan amyE::P$_{hyper}$-YfmR | This study |
| HAF527 | 168 trpC2 Δefp ΔyebC2::kan amyE::P$_{hyper}$-YebC2 | This study |
| HAF528 | 168 trpC2 Δefp ΔyfmR amyE::P$_{hyper}$-YebC2 | This study |
| HRH774 | 168 trpC2 WT lacA::P$_{xyl}$-dCas9 | This study |
| HRH776 | 168 trpC2 Δefp::kan lacA::P$_{xyl}$-dCas9 | This study |
| HRH1022 | 168 trpC2 ΔyfmR::kan lacA::P$_{xyl}$-dCas9 | This study |
| HRH1024 | 168 trpC2 ΔyebC2::kan lacA::P$_{xyl}$-dCas9 | This study |
| HRH1134 | 168 trpC2 ΔyebC2::kan ΔyfmR lacA::P$_{xyl}$-dCas9 | This study |
| HRH829 | 168 trpC2 Δefp::kan lacA::P$_{xyl}$-dCas9 amyE::P$_{veg}$-sgRNA$^{yebC2}$ | This study |
| HRH1042 | 168 trpC2 ΔyfmR::kan lacA::P$_{xyl}$-dCas9 amyE::P$_{veg}$-sgRNA$^{efp}$ | This study |
| HRH1053 | 168 trpC2 ΔyebC2::kan lacA::P$_{xyl}$-dCas9 amyE::P$_{veg}$-sgRNA$^{efp}$ | This study |
| HRH1137 | 168 trpC2 ΔyebC2::kan ΔyfmR lacA::P$_{xyl}$-dCas9 amyE::P$_{veg}$-sgRNA$^{efp}$ | This study |
| HRH1177 | 168 trpC2 WT sacA::P$_{hyper}$-3xFLAG-*rfp-cfp* | This study |
| HRH1193 | 168 trpC2 Δefp sacA::P$_{hyper}$-3xFLAG-*rfp-cfp* | This study |
| HRH1178 | 168 trpC2 ΔyfmR sacA::P$_{hyper}$-3xFLAG-*rfp-cfp* | This study |
| HRH1179 | 168 trpC2 ΔyebC2 sacA::P$_{hyper}$-3xFLAG-*rfp-cfp* | This study |
| HRH1195 | 168 trpC2 Δefp ΔyfmR::kan sacA::P$_{hyper}$-3xFLAG-*rfp-cfp* | This study |
| HRH1197 | 168 trpC2 Δefp ΔyebC2 sacA::P$_{hyper}$-3xFLAG-*rfp-cfp* | This study |
| HRH1180 | 168 trpC2 ΔyebC2 ΔyfmR sacA::P$_{hyper}$-3xFLAG-*rfp-cfp* | This study |
| HRH1185 | 168 trpC2 Δefp amyE::P$_{hyper}$-YfmR sacA::P$_{hyper}$-3xFLAG-*rfp-cfp* | This study |
| HRH1187 | 168 trpC2 Δefp amyE::P$_{hyper}$-YebC2 sacA::P$_{hyper}$-3xFLAG-*rfp-cfp* | This study |
| HRH1199 | 168 trpC2 Δefp ΔyfmR::kan amyE::P$_{hyper}$-YebC2 sacA::P$_{hyper}$-3xFLAG-*rfp-cfp* | This study |
| HRH1201 | 168 trpC2 Δefp ΔyebC2::kan amyE::P$_{hyper}$-YfmR sacA::P$_{hyper}$-3xFLAG-*rfp-cfp* | This study |
| HRH1203 | 168 trpC2 Δefp ΔyebC2::kan amyE::P$_{hyper}$-YebC2 sacA::P$_{hyper}$-3xFLAG-*rfp-cfp* | This study |
| HRH1205 | 168 trpC2 Δefp ΔyfmR::kan amyE::P$_{hyper}$-YfmR sacA::P$_{hyper}$-3xFLAG-*rfp-cfp* | This study |
| HRH1181 | 168 trpC2 WT sacA::P$_{hyper}$-3xFLAG-*rfp-5xprolines-cfp* | This study |
| HRH1194 | 168 trpC2 Δefp sacA::P$_{hyper}$-3xFLAG-*rfp-5xprolines-cfp* | This study |
| HRH1182 | 168 trpC2 ΔyfmR sacA::P$_{hyper}$-3xFLAG-*rfp-5xprolines-cfp* | This study |
| HRH1183 | 168 trpC2 ΔyebC2 sacA::P$_{hyper}$-3xFLAG-*rfp-5xprolines-cfp* | This study |
| HRH1207 | 168 trpC2 Δefp ΔyfmR::kan sacA::P$_{hyper}$-3xFLAG-*rfp-5xprolines-cfp* | This study |
| HRH1209 | 168 trpC2 Δefp ΔyebC2::kan sacA::P$_{hyper}$-3xFLAG-*rfp-5xprolines-cfp* | This study |
| HRH1184 | 168 trpC2 ΔyebC2 ΔyfmR sacA::P$_{hyper}$-3xFLAG-*rfp-5xprolines-cfp* | This study |
| HRH1190 | 168 trpC2 Δefp amyE::P$_{hyper}$-YfmR sacA::P$_{hyper}$-3xFLAG-*rfp-5xprolines-cfp* | This study |

*(Continued)*

**Table 1.**  (Continued)

| Strain (strain number) | Description | Source |
|---|---|---|
| HRH1191 | 168 trpC2 Δefp amyE::P$_{hyper}$-YebC2 sacA::P$_{hyper}$-3xFLAG-rfp-5xprolines-cfp | This study |
| HRH1211 | 168 trpC2 Δefp ΔyfmR::kan amyE::P$_{hyper}$-YebC2 sacA::P$_{hyper}$-3xFLAG-rfp-5xprolines-cfp | This study |
| HRH1213 | 168 trpC2 Δefp ΔyebC2::kan amyE::P$_{hyper}$-YfmR sacA::P$_{hyper}$-3xFLAG-rfp-5xprolines-cfp | This study |
| HRH1215 | 168 trpC2 Δefp ΔyebC2::kan amyE::P$_{hyper}$-YebC2 sacA::P$_{hyper}$-3xFLAG-rfp-5xprolines-cfp | This study |
| HRH1217 | 168 trpC2 Δefp ΔyfmR::kan amyE::P$_{hyper}$-YfmR sacA::P$_{hyper}$-3xFLAG-rfp-5xprolines-cfp | This study |
| HAF602 | 168 trpC2 Δefp ΔyebC2::kan amyE::P$_{hyper}$-yebC2$^{yrbC-D1}$ sacA::P$_{hyper}$-3xFLAG-rfp-5xprolines-cfp | This study |
| HF603 | 168 trpC2 Δefp ΔyebC2::kan amyE::P$_{hyper}$-yrbC$^{yebC2-D1}$ sacA::P$_{hyper}$-3xFLAG-rfp-5xprolines-cfp | This study |
| HAF604 | 168 trpC2 Δefp ΔyebC2::kan amyE::P$_{hyper}$-yrbC sacA::P$_{hyper}$-3xFLAG-rfp-5xprolines-cf | This study |
| HAF586 | 168 trpC2 Δefp amyE::P$_{hyper}$-yrbC sacA::P$_{hyper}$-3xFLAG-rfp-5xprolines-cfp | This study |
| HAF582 | 168 trpC2 Δefp ΔyrbC::kan sacA::P$_{hyper}$-3xFLAG-rfp-5xprolines-cfp | This study |
| **Plasmid** | **Description** | **Source** |
| pHRH703 | pDR111 amyE::P$_{hyper}$-$^{B.}$ subtilis YebC2 | This study |
| pHRH706 | pDR111 amyE::P$_{hyper}$-$^{B.}$ subtilis YfmR | [20] |
| pHRH1021 | pJMP2 amyE::P$_{veg}$-sgRNA$^{efp}$ | This study |
| pHRH819 | pJMP2 amyE::P$_{veg}$-sgRNA$^{yebC2}$ | This study |
| pHRH899 | pDR111 amyE::P$_{hyper}$-3xFLAG-rfp-cfp | [20] |
| pHRH903 | pDR111 amyE::P$_{hyper}$-3xFLAG-rfp-5xprolines-cfp | [20] |
| pECE174 | SacA integration plasmid to $^{B.}$ subtilis | [62] |
| pHRH1169 | pECE174 sacA::P$_{hyper}$-3xFLAG-rfp-cfp | This study |
| pHRH1173 | pECE174 sacA::P$_{hyper}$-3xFLAG-rfp-5xprolines-cfp | This study |
| **Primer** | **Sequence** | **Source** |
| HRH sgRNA-efp-3 | 5'- gctcgtgttgtacaataaatgtatcgcgccagtgcgaaggttggtttttagagctagaaatagcaagt-taaaataaggc -3' | This study |
| HRH sgRNA-YeeI-3 | 5'- gctcgtgttgtacaataaatgtacgccgccacataaatctcagttttagagctagaaatagcaagt-taaaataaggc -3' | This study |
| HRH175 | 5'- acatttattgtacaacacgagcc-3' | [20] |
| HRH155 | 5'- taattgtgagcggataacaattaagcttggaggaaaaaaaatgggccgtaagtggaaca -3' | This study |
| HRH156 | 5'- ctcgtttccaccgaattagcttgcatgcttactcacctaaatcaacgttatgatatacc -3' | This study |
| HRH157 | 5'- attgtgagcggataacaattaagcttggaggaaaaaaaatgagcatattaaaagcggaa -3' | This study |
| HRH158 | 5'- acctcgtttccaccgaattagcttgcatgcttagctttccagttcttcga -3' | This study |
| HRH204 | 5'- gccgatgataagctgtcaaacatgagaattcgactctctagcttgaggcatc -3' | This study |
| HRH205 | 5'- tggtaatggtagcgaccggcgctcaggatcctaactcacattaattgcgttgc -3' | This study |

plasmids, pHRH703 (P$_{hyper}$-YebC2) and pHRH706 (P$_{hyper}$-YfmR), were linearized with ScaI and transformed for integration on the chromosome at *amyE*. For the experiment exchanging Domain I of YebC2 and YrbC residues 1–74 of YebC2 were exchanged with residues 1–76 of YrbC. The chimeric versions of these genes were ordered as gene blocks from Integrated DNA Technologies and assembled into pDR111 by Gibson assembly. For over-expression of YrbC and YebC2, untagged *yrbC* and *yebC2* were ordered as gene blocks from Integrated DNA Technologies and assembled into pDR111 by Gibson assembly.

## Growth curves

*B. subtilis* strains were grown overnight at room temperature, and inoculated to a final $OD_{600}$ 0.05 in 150 μl LB, and supplemented with 1 mM IPTG where appropriate in a 96 well-plate (ThermoScientific 167008). The cultures were incubated at 30°C and 37°C with linear shaking (2-mm intensity). $OD_{600}$ of strains was measured at 15-minute intervals over 20 hours using a microplate reader (BioTek).

## Colony size measurement

*B. subtilis* strains were cultured in LB at room temperature or 37°C overnight in a roller drum at 80 rpm. 1 mM IPTG was added to the strains overexpressing Yebc2 or YfmR. The cells were normalized to $OD_{600}$ 0.05, serially diluted, and plated onto two LB agar plates for incubation at 30°C or 37°C for 24 hours, and placed at room temperature for 24 hours before imaging with ChemiDoc MP (Biorad). The area of individual colonies was quantified using ImageJ [64].

## CRISPRi depletion

Primer HRH sgRNA-efp-3 containing an sgRNA sequence (5'-tcgcgccagtgcgaaggttg-3') was designed to target EF-P. HRH sgRNA-efp-3 and HRH175 [20] were used to amplify pJMP2 [40], generating pHRH1021. pJMP1 carrying dCas9 under a xylose-inducible promoter [40] was transformed into the single deletion strains *ΔyebC2::kan* (HAF450) and *ΔyfmR::kan* (HAF451) and the double deletion strain *ΔyebC2ΔyfmR* (HRH1132). Next, pHRH1021 was transformed into the strains harboring dCas9, therefore producing EF-P depletion strains. The resulting *B. subtilis* CRISPRi knockdown strains were cultured overnight without xylose and diluted to an $OD_{600}$ 0.05 in PBS. The cultures were subsequently diluted 10-fold as $10^{-2}$ to $10^{-6}$ and spotted onto LB agar without xylose or onto LB agar containing 5% xylose and incubated for 12 hours at 37˚C.

## Proline stalling reporter and western blots

The RFP-CFP fusion cassette containing a pentaproline stalling motif (5'-ccaccaccac caccc-3') or the reporter cassette without the motif were amplified using primers HRH204 and HRH205 from the previous constructs pHRH899 and pHRH903 (Table 1). The resulting fragments were cloned into pECE174 [62] plasmid cut with EcoRI and BamHI, producing pHRH1169 and pHRH1173. The resulting reporter plasmids were sequenced by Plasmidsaurus and linearized with ScaI to transform into the different combinations of deletions in *B. subtilis* for recombination at *sacA*. The reporter strains were grown overnight with 1 mM IPTG and then diluted back to $OD_{600}$ 0.05. The diluted cultures were induced with 1 mM IPTG and grown up to $OD_{600}$ 1.2 at 37°C. Cell cultures were normalized by OD and resuspended in 60 μL of lysis buffer (10 mM Tris pH 8, 50 mM EDTA, 1 mg/mL lysozyme), then incubated at 37°C for 10 min then added to 4x SDS-PAGE loading buffer. Samples were heated at 85 °C for 5 min and immediately cooled on ice. 12 μL samples were loaded onto a 12% SDS-PAGE gel and run at 150 V for 70 min. The protein was transferred to PVDF membrane (Biorad) at 300 mAmp for 110 min. The membrane was blocked with 3% BSA for 20 min and incubated with 1 μL anti-FLAG monoclonal antibody (Sigma A8592) in 10 mL 3% BSA for 2 hours at room temperature. The membrane was washed 3 times with PBS-T and developed with ECL (Biorad170-5060) for 2 min and imaged on ChemiDocMP (Biorad).

## Polysome profiling

Strains were grown overnight at 37°C and inoculated to an $OD_{600}$ of 0.05 in 40 ml LB the next morning. Cells were collected at $OD_{600}$ 1.2 by centrifugation at 8000 rpm for 10 minutes (Beckman Coulter Avanti J-15R, rotor JA-10.100). Cell pellets were resuspended in 200 μl gradient buffer containing 20 mM Tris (pH 7.4 at 4°C), 0.5 mM EDTA, 60 mM $NH_4Cl$, and 7.5 mM $MgCl_2$ and 6mM 2-mercaptoethanol. Cells were lysed using a homogenizer (Bead-bug6, Benchmark) by five 20 second pulses at speed 4350 rpm with chilling on ice for 2 min between the cycles and clarified by centrifugation at 21,300 rcf for 20 min (Eppendorf 5425R, rotor FA-24x2). Clarified lysates were normalized to 1500 ng/μl and loaded onto 10–40% sucrose gradients in gradient buffer and run for 3 hours at 30,000 rpm at 4°C in an SW-41Ti rotor. Gradients were collected using a Biocomp Gradient Station (BioComp Instruments) with A260 continuous readings (Triax full spectrum flow cell). The area under each peak was quantified using Graphpad Prism.

## Detection of YebC2 ribosome association

We constructed a strain expressing YebC2 encoding an in-frame 6X-Histidine tag immediately after Gly74 in an unstructured loop. We verified that this tagged YebC2 was functional by determining that it could complement the Δ*efp*Δ*yebC2* growth defect (S2 Fig). Ribosomes from this strain were resolved by sucrose density gradient ultracentrifugation as described for polysome profiling. Resulting fractions were resolved by SDS-PAGE, transferred to PVDF membrane and probed with anti-His antibody (Invitrogen MA1-21315-HRP).

## Gene detection

Genes were detected in a database of >18,000 representative prokaryotic genomes from NCBI RefSeq using HMMER v3.3 (nhmmer) (hmmer.org) with an E-value cutoff of 0.05 and a query of characterized *yebC*-family gene sequences (S1 Data). Hits were classified as either *yebC* or *yebC2* depending on the gene query that resulted in a higher sequence bit score, and therefore greater homology. Genomes were filtered for <10% CheckM contamination [65], which left us with 15,259 genomes to survey.

## Phylogenetics and protein modeling

16S rRNA sequences of all genomes were identified and acquired using BLAST v2.13.0, aligned using MAFFT v7.453, and applied to FastTree v2.1.11 [66] to infer a maximum-likelihood tree [67]. FastTree produces unrooted phylogenies, so the tree was midpoint rooted using the phangorn v2.11.1 package [68]. Taxonomic classification was assigned to genomes using the NCBI Taxonomy database [69] and taxonkit v0.17.0 [70]. Phyla were named using the conventions in Coleman et al. 2021 [71]. The tree was visualized using ggtree v3.12.0 [72]. For the large YebC-family tree, the gene sequences of HMMER hits were translated using a Python script, and the tree was built and visualized with iTol v7.0 [73]. Protein models were determined with AlphaFold [74,75] and visualized with ChimeraX version 1.8 [43].

## Supporting information

**S1 Fig. YebC2 over-expression improves fitness of Δ*efp*Δ*yfmR* cells.** (Left) Growth in LB liquid media at 37°C of wild-type (WT), Δ*efp*, Δ*yfmR*, Δ*efp*Δ*yfmR* and Δ*efp*Δ*yfmR* cells expressing IPTG-inducible YebC2. **(Right)** Colony sizes on LB plates of various mutants after 24 hours of growth at 37°C. YebC2 or YfmR was expressed from an IPTG-inducible promoter.

Error bars represent standard deviation. P-vaules report the result of an unpaired *t*-test with Welch's correction.
(TIF)

**S2 Fig.  His-tagged YebC2 is functional and complements the growth defect of Δ*efp*Δ*yebC2* cells *in vivo*.** Growth rates in LB at 37˚C are shown for wild-type, Δ*yebC2*, Δ*efp*, Δ*efp*Δ*yebC2* and Δ*efp*Δ*yebC2* expressing His-tagged YebC2. Error bars represent standard deviation of two independent experiments.
(TIF)

**S3 Fig.  Midpoint rooted maximum-likelihood tree and sequence similarity matrix of characterized YebC family proteins** . Characterized YebC family proteins are labelled with their given gene name and respective organism*: Bs, Bacillus subtilis; Ld, Lactobacillus delbrueckii; Pa, Pseudomonas aeruginosa; Ec, Escherichia coli; Bb, Borrelia burgdorferi; Sp, Streptococcus pyogenes*. Maximum likelihood bootstrap values are listed at each node. Pairwise percent identities for the proteins are listed and shaded relative to their homology.
(TIF)

**S4 Fig.  Multisequence alignment for a selection of YebC2 and YebC paralogs.** Amino acid sequences were aligned with Clustal Omega. YebC2 paralogs are shaded in gray and YebC paralogs are shaded in green. Species abbreviations: *Bs, Bacillus subtilis; Ec, Escherichia coli; Sp, Streptococcus pyogenes.*
(TIF)

**S1 Table.  A table of taxa used to build the phylogenetic tree in** Fig 6 **and the number of** *yebC* **and** *yebC2* **genes identified in each taxa.**
(XLSX)

**S1 Data.  Primary data underlying graphs in figures.**
(XLSX)

## Acknowledgements

We are grateful to Tory Hendry, Vasili Hauryliuk, and Allen Buskirk for feedback on the manuscript. We are grateful to Kevin England, and Daniel Tetreault for helpful discussion.

## Author contributions

**Conceptualization:** Heather Feaga.

**Data curation:** Hye-Rim Hong, Cassidy R. Prince, Letian Wu, Heather Feaga.

**Formal analysis:** Hye-Rim Hong, Cassidy R. Prince, Letian Wu, Isabella N. Lin, Katrina Callan, Heather Feaga.

**Funding acquisition:** Heather Feaga.

**Investigation:** Hye-Rim Hong, Cassidy R. Prince, Letian Wu, Isabella N. Lin, Katrina Callan, Heather Feaga.

**Methodology:** Hye-Rim Hong, Cassidy R. Prince, Letian Wu, Heather Feaga.

**Project administration:** Heather Feaga.

**Supervision:** Heather Feaga.

**Visualization:** Letian Wu.

**Writing – original draft:** Hye-Rim Hong, Cassidy R. Prince, Heather Feaga.

**Writing – review & editing:** Hye-Rim Hong, Cassidy R. Prince, Letian Wu, Katrina Callan, Heather Feaga.

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
