## [Decision Letter · Decision Letter 0]

2 Dec 2024

PGENETICS-D-24-01273

YebC2 resolves ribosome stalling independent of EF-P and the ABCF ATPase YfmR

PLOS Genetics

Dear Dr. Feaga,

Thank you for submitting your manuscript to PLOS Genetics. After careful consideration, we feel that it has merit but does not fully meet PLOS Genetics's publication criteria as it currently stands. Therefore, we invite you to submit a revised version of the manuscript that addresses the points raised during the review process.

Please submit your revised manuscript within 60 days Jan 31 2025 11:59PM. If you will need more time than this to complete your revisions, please reply to this message or contact the journal office at plosgenetics@plos.org. Please include the following items when submitting your revised manuscript:

We look forward to receiving your revised manuscript.

Kind regards,

Shumin Tan, Ph.D.

Academic Editor

PLOS Genetics

Sean Crosson

Section Editor

PLOS Genetics

Aimée Dudley

Editor-in-Chief

PLOS Genetics

Anne Goriely

Editor-in-Chief

PLOS Genetics

**Additional Editor Comments :**

While there was appreciation for the interesting data presented, more robust support for some of the conclusions stated, such as those regarding the independent action of YebC2 from EF-P and YfmR, the impact of YebC2 association with 70S polysomes, and differences between YebC and YebC2, are needed to provide the improved mechanistic detail required for PLoS Genetics.

**Journal Requirements:**

- TM on pages: 17, and 18.

4) We notice that your supplementary Figures are included in the manuscript file. Please remove them and upload them with the file type 'Supporting Information'.  Please add a full list of legends for your Supporting Information files after the references list.

5) We note that your Data Availability Statement is currently as follows: "Data is available in supplemental files Table S1 and Table S2.". Please confirm at this time whether or not your submission contains all raw data required to replicate the results of your study. Authors must share the “minimal data set” for their submission. PLOS defines the minimal data set to consist of the data required to replicate all study findings reported in the article, as well as related metadata and methods (https://journals.plos.org/plosone/s/data-availability#loc-minimal-data-set-definition).

2) State what role the funders took in the study. If the funders had no role in your study, please state: "The funders had no role in study design, data collection and analysis, decision to publish, or preparation of the manuscript.".

If you did not receive any funding for this study, please simply state: u201cThe authors received no specific funding for this work.

**Reviewers' comments:**

Reviewer's Responses to Questions

Reviewer #1: Hong and colleagues characterise a novel translation factor, YevbC2, and show that the factor is involved in resolving ribosomal stalls on polyproline tracts. The results are interesting and convincing, although somewhat limited in mechanistic detail. However, as this study is one of the first reports describing the biological function of the factor, the level of detail is adequate. I have several comments / suggestions.

Throughout the text, including figure legends, the authors refer to ‘severe growth and fitness defects’. ‘Severe’ is too strong a word here, I feel. I would use something less dramatic.

Line 58: 'The Escherichia coli homolog of YfmR, Uup'. I think here one could be more specific and say ‘orthologue’ – while Uup is, indeed, a homologue of YfmR, so are the other ABCF ATPases. However, the connection between Uup and YfmR is tighter than that.

YebC2 over-expression rescues the synthetic fitness defect of ∆efp∆yfmR: could you please also show serial dilutions data in addition to colony size measurements? Or growth rate measurements? colony size measurements are a rather exotic way to assess fitness.

Have the authors attempted generating a ∆efp∆yebC2 double KO strain?

Line 189: 'YebC2 associates with 70S ribosomes'. Polysome analysis shows that YebC2 signal is present in 30S, 50S and 70S fractions, but not in polysomes. I would be careful and not make strong concussions about 70S being the target. 30S and 50S signal is similarly strong. Moreover, there is no signal on the polysomes… Obviously, the factor might not interact stably with the target ribosomal complex and be lost during the centrifugation. Please reword ‘We found that YebC2 co-migrates with ribosomes and was strongly associated with 70S ribosomes’ and highlight the lack of the interaction with polysomes.

Figure 4: while bootstraps are mentioned in the text, I somehow miss them on the figure / figure legend. Please revise for clarity so they are more obvious. I would compare the structures (experimental or AF2-generated) for YebC2 and YebC: what is the structural difference? What makes one a translation factor and the other a transcription factor?

Figure 5: please italicise yebC2 gene.

Line 277, 'EfpLhave'. A space is missing.

Line 289: 'Helix 89' – please specify if it is a small or a large subunit, and where on the subunit the helix is located. Not everybody is familiar with ribosomal landmarks.

Line 49 and 298: 'P-site' should be P site. The hyphen would be needed when we have a compound adjective, like 'P-site tRNA', line 306. When we are referring to the P site as such, no hyphen is needed. Same applies to line 287, 290, 309 ‘A-site’.

Line 33, ‘E.coli’. A space is missing.

Reviewer #2: The manuscript by Hong et al, describes the role of YebC2 (YeeI) in preventing ribosome stalling in Bacillus subtilis. Using clever and clear genetics they show that depletion of YebC2 impairs viability of cells lacking Efp and that the absence of YebC2 increases the rate of ribosome stalling and abortive translation. Finally, they show that YebC2 associates with ribosomes and promotes 70S ribosome and polysome production. They round out the characterization with nice phylogenetic analyses. The work is interesting and sound. I only have a few minor comments.

Line 124. The ymfR yebC2 double mutant comes a bit out of nowhere here. What is the phenotype of the ymfR yebC2 double? Maybe just start with depleting efp in the single mutants and describe relative fitness before moving on to the double. Starting the paragraph with the undescribed double creates a massive leap in logic that was distracting/confusing. I note that what I described above is also consistent with the order of strains in figure 2A.

Line 166. What is known about truncated peptide production in B. subtilis? Is this like the ssrA trans-translation system? And why is the truncated product not destroyed by proteolysis? In general, this statement needs better support/explanation.

Figure 4 and results. What distinguishes YebC from YebC2? Are there particular domains added/missing? Does YebC2 lack the HTH or whatever promotes DNA binding of the YebC subfamily? A multiple sequence alignment comparing YebC to YebC2 proteins in prominent model organisms might be useful to see at a glance how/where the subfamilies differ.

Phylogenetic distribution. Any chance that the presence of YebC2 has anything to do with rapid growth rates?

Line 289. What is Helix 89?

Reviewer #3: This manuscript describes the identification of YebC2 as a translation factor of B. subtilis that resolves ribosome stalling at poly-Pro sequences. The overall conclusion is not that novel, because (1) the homologs of the protein in human and in S. pyogenes have been demonstrated with such role, and (2) the protein was identified from the same screen that the authors published previously for proteins that can rescue the efp-KO defect, knowing that efp is the main factor that resolves ribosome stalling at poly-Pro sequences. The lack of novelty of the overall conclusion is compounded by the incomplete analysis of the YebC2 protein in multiple places, where the authors simply stated that “further experiments are necessary”. In fact, these are exactly the experiments that are necessary to convincingly demonstrate the proposed role of YebC2. Below both major and minor weaknesses of the manuscript are listed.

Conceptual weaknesses:

1. The authors stated that YebC2 is “independent” of EF-P and YfmR as a factor that resolves ribosome stalling at poly-Pro sequences. However, “independent” needs to be quantitatively demonstrated by measuring the effect of each protein. If EF-P-KO has a growth defect of x-fold, YfmR-KO has an effect of y-fold, and YebC2-KO has an effect of z-fold, these fold changes need to be directly compared. It is a mis-opportunity that the data in Figure 2B is not more carefully analyzed. Even if Yeb2C has a rescue effect in the strain of EF-P-KO/YfmR-KO, this does not mean that Yeb2DC is independent of the two other genes. Until further analysis, the claim of an ”independent” effect is not justified.

2. Just because YebC2 is associated with the 70S, it does not mean that it facilitates translation of the ribosome. It may provide a quality control that regulates the ribosome activity. Additionally, it appears that YebC2-KO leads to accumulation of 50S and 30S, indicating that it prevents the 70S assembly. An explanation of this role is completely absent from the manuscript.

3. While phylogenetic analysis showed that YebC2 is separated from other members of the YebC family, this does not mean it has only a function in translation and not in transcription as in other members of the YebC family. The example is the E. coli YebC2 enzyme, which appears to have a role in both. The conclusion made by the authors is again very superficial and without experimental backing.

Technical weaknesses:

1. The authors used the word “more essential” multiple times in the text. In Genetics, “essential” is a definitive word – either it is required or not required for growth. The phrase “more essential” is unclear.

2. Table 1 can go to the Supplement.

3. Figure 2A, the Y axis is unclear. It should be #generations per hour.

4. The writing in line 130 is very difficult to read. The authors wanted to say “when compared to when EF-P was knocked out alone”?

5. The data in Figure 2A of the spot test do not convey the results. The most critical presentation should be the effect of EF-P-KO. This figure should be re-done.

6. In contrast to authors’ claim, there is no effect of yebC2 complementation at 37 oC in Figure 2B. The authors incorrectly interpreted the results?

7. Line 174. The “additive increase in ribosome stalling for the …..”, the word “additive” needs to be justified. Without a quantitative measurement, this word was incorrectly used.

8. Figure 3A needs a control of the reporter gene, not a poly-Pro sequence. In the same figure, “stalled” is mis-leading. Instead, “truncated peptide” would be more appropriate.

9. Figure 3B, where does YebC2 bind to the ribosome? It appears to bind 70S, 50S, and 30S. What is the meaning of this binding? In the same figure, a control of EF-P is necessary, but lacking.

10. Evidence that YebC2 binding to the A site should be directly demonstrated. This is lacking and thus the proposed model is incomplete.

11. A simple prediction of YebC2 by an Alpha-Fold modeling should have been done to predict its structure. This is lacking, making the proposed model incomplete.

**Have all data underlying the figures and results presented in the manuscript been provided?**

Reviewer #1: None

Reviewer #2: Yes

Reviewer #3: **No: **Data for YebC2 binding to the A site, and independent of EF-P-KO/YrmR-KO need to be presented. The data in Fig 2A do not provide the direct information, no support for YebC2 to have a role at 37 oC (Fig. B). A quantitative measurement to support an "additive" effect is lacking. Controls for YebC2 rescuing the ribosome specifically for poly-Pro are absent, and an EF-P control for direct binding to the ribosome is lacking. Evidence for YebC2 binding to the A site, or binding in a way distinct from EF-P binding of YfmR binding to the ribosome is lacking.

PLOS authors have the option to publish the peer review history of their article (what does this mean?). If published, this will include your full peer review and any attached files.

Reviewer #1: No

Reviewer #2: No

Reviewer #3: No

**Figure resubmission:**
---

## [Decision Letter · Decision Letter 1]

22 Feb 2025

Dear Dr Feaga,

We are pleased to inform you that your manuscript entitled "YebC2 resolves ribosome stalling and increases fitness of cells lacking EF-P and the ABCF ATPase YfmR" has been editorially accepted for publication in PLOS Genetics. Congratulations!

Yours sincerely,

Shumin Tan, Ph.D.

Academic Editor

PLOS Genetics

Sean Crosson

Section Editor

PLOS Genetics

Aimée Dudley

Editor-in-Chief

PLOS Genetics

Anne Goriely

Editor-in-Chief

PLOS Genetics

Comments from the reviewers (if applicable):

Reviewer's Responses to Questions

**Comments to the Authors:**

Reviewer #1: I am happy with the revison!

Reviewer #2: The authors have addressed my concerns.

Reviewer #3: The use of the "stalled" protein is still not appropriate. Just because it is used in the field, does not mean that it is the best use of word. Revise it to "protein synthesis that is stalled", which would be better.

**Have all data underlying the figures and results presented in the manuscript been provided?**

Reviewer #1: Yes

Reviewer #2: Yes

Reviewer #3: Yes

PLOS authors have the option to publish the peer review history of their article (what does this mean?). If published, this will include your full peer review and any attached files.

Reviewer #1: No

Reviewer #2: No

Reviewer #3: No

**Data Deposition**

http://datadryad.org/submit?journalID=pgenetics&manu=PGENETICS-D-24-01273R1

**Press Queries**

---

## [Editor Report · Acceptance letter]

PGENETICS-D-24-01273R1

YebC2 resolves ribosome stalling and increases fitness of cells lacking EF-P and the ABCF ATPase YfmR

Dear Dr Feaga,

We are pleased to inform you that your manuscript entitled "YebC2 resolves ribosome stalling and increases fitness of cells lacking EF-P and the ABCF ATPase YfmR" has been formally accepted for publication in PLOS Genetics! Your manuscript is now with our production department and you will be notified of the publication date in due course.

With kind regards,

Anita Estes

PLOS Genetics

On behalf of:
